# The δ Phase Precipitation of an Inconel 718 Superalloy Fabricated by Electromagnetic Stirring Assisted Laser Solid Forming

**DOI:** 10.3390/ma12162604

**Published:** 2019-08-15

**Authors:** Feiyue Lyu, Fencheng Liu, Xiaoan Hu, Xiaoguang Yang, Chunping Huang, Duoqi Shi

**Affiliations:** 1School of Aeronautical Manufacturing and Engineering, Nanchang Hangkong University, Nanchang 330063, China; 2School of Aircraft Engineering, Nanchang Hangkong University, Nanchang 330063, China; 3School of Energy and Power Engineering, Beihang University, Beijing 100191, China

**Keywords:** electromagnetic stirring, laser solid forming, laves phase, δ phase, precipitation behavior

## Abstract

Fabricating an Inconel 718 superalloy using electromagnetic stirring assisted laser solid forming (EMS-LSF) is a novel method to modify its microstructure and mechanical properties by consuming the Nb element in the γ phase to alleviate interdendritic segregation. The precipitate of the δ phase at 950 °C after EMS-LSF can help to achieve the uniform diffusion of Nb, and can also improve its mechanical properties. The precipitation behavior of the δ phase in an EMS-LSF Inconel 718 superalloy with different heat treatment processes has been investigated. The results show that the morphology of the δ phase changes from rod-like to a long-needle shape and tends to grow from the inter dendrite to the core dendrite with electromagnetic field intensity increasing, which is accompanied by the “cutting” and “dissolution” of the Laves phase. Through precipitation kinetics analysis, the precipitation rate of the δ phase is seen to increase with the electromagnetic field intensity increasing. Under a combination of electromagnetic stirring and laser solid forming, the microhardness of the Inconel 718 samples increased slightly due to the fact that a higher content of Nb was distributed in the core dendrite resulting from the serious convection of liquid metal, which can strengthen the matrix.

## 1. Introduction

The Inconel 718 superalloy is a Ni-Fe-Cr based and precipitation-hardening superalloy. It is widely used in various kinds of disks, rotors, shafts, fasteners, and other components because of its excellent strength, good corrosion resistance, and favorable weldability [1].

Laser solid forming (LSF) is a novel near-net-shape manufacturing process for metals, which can fabricate complex structures freely and rapidly [2]. Compared to traditional manufacturing methods, such as casting and forging processes, LSF yields unique benefits and advantages, including high flexibility, low material consumption, and reduction of production steps [3]. So far, it has been proven that LSFed Inconel 718 parts have superior performance, such as in their tensile properties, compared to forged components [2].

At present, studies on LSFed Inconel 718 have mainly focused on its forming process, microstructure, and mechanical properties [4,5,6,7,8,9,10,11]. It can be found that Nb is highly susceptible to segregation and leads to the precipitation of some brittle phases, such as the Laves phase in the LSFed Inconel 718 superalloy, known for degrading the tensile ductility, impact properties, fatigue properties, and plasticity of materials [2,4,5,12]. Therefore, there have been few applications using the LSF technique to manufacture superalloy structures in recent years.

Indeed, some researchers have already found that solid solution treatment with a traditional high temperature (1150 °C) can eliminate interdendritic segregation and reduce the brittle Laves phase. Meanwhile, it can also cause grain coarsening in materials and lead to its mechanical properties declining sharply [12]. Liu et al. [13] have also found that some heat treatments may alleviate interdendritic segregation to eliminate the Laves phase with a high solid-solution temperature or a long aging time. However, they obtain a coarse and unevenly distributed grain structure, which is bad for the fatigue and other properties of LSFed Inconel 718. In order to solve this problem and improve the mechanical properties of the material, a new heat treatment called the δ aging treatment has been proposed. Chen et al. [14] found an effective way to alleviate the segregation of elements existing in the inter dendrite in the Inconel 718 alloy and reduce the Laves phase after δ aging treatment between 890 °C and 1000 °C. This means that the δ aging treatment can alleviate segregation and reduce the brittle Laves phase by precipitating the δ phase in a lower temperature, which is good for the material. Song et al. [15,16] explored the dislocation pile up at the δ phase owing to the larger size of the δ phase in the heat treated Inconel 718 alloy compared with that in the wrought alloy; δ phase can improve the alloy’s mechanical properties and easily reduce the notch sensitivity of the alloy. Slama et al. [17,18] have described how the precipitation and elongation of the δ phase through the consuming γ″ phase occurs near the grain boundaries after heat treatment; this plays an important role in controlling grain growth to improve mechanical properties. The δ phase is a stable phase with an orthorhombic structure, and its molecular formula is Ni_3_(Nb_0.8_Ti_0.2_). The γ” phase is a metastable phase of the body-centered tetragonal structure, which belongs to the Ni_3_Nb phase. Radavich et al. [19,20] have also reported that the δ phase precipitated at the grain boundary and formed a needle-liked δ phase with an extension of the aging time, thereby reinforcing the material in a temperature range of 1100–1350 °C.

However, δ aging treatment cannot alleviate the segregation of elements and eliminate the Laves phase completely in inter dendrite components [14]. Therefore, it is beneficial to combine electromagnetic stirring (EMS) and LSF to further homogenize the alloying elements. So far, EMS has been widely used in the welding field, as EMS can obviously reduce the number of pores, inhibit the production of columnar crystals, and improve the microhardness and the mechanical properties in aluminium alloy, stainless steel, magnesium alloys and titanium alloys [21,22,23,24]. In addition, it also has obvious effects on reducing defects, refining grains, improving the homogeneity of the microstructure and reducing the residual stress of specimens in the additive manufacturing field [25].

There are plenty of documents [26,27,28] discussing the influence of the electromagnetic field on the microstructure of the LSFed Inconel 718 alloy. These documents reveal the mechanisms behind grain refinement. The effect of linear electromagnetic field on the microstructure of superalloys has also been reported. The precipitation of the δ phase plays an important role in improving the mechanical properties of the alloy, but there is a lack of detailed investigation on the precipitation of the δ phase to homogenize the alloying elements via the method of electromagnetic stirring to improve the mechanical properties in the process of laser solid forming.

In this paper, electromagnetic stirring was introduced to a LSFed Inconel 718 superalloy to adjust and control the volume fraction, morphology, and distribution of the δ phase to mitigate the microsegregation of the alloying elements. Meanwhile, the precipitation behavior of the δ phase was investigated in EMS-LSF, which aimed to homogenize the elements to further improve the mechanical properties of the LSFed Inconel 718 superalloy through different aging times.

## 2. Experimental Methods

The substrate used in this experiment was 304 stainless steel sheets with dimensions of 150 mm × 80 mm × 10 mm. The test block of the Inconel 718 superalloy was formed by laser solid forming on a plane of 150 mm × 80 mm. The surface of the substrate was polished with sandpaper, to remove surface oxidation, and was cleaned with acetone before the laser solid forming (LSF) process.

The deposition material was an Inconel 718 superalloy powder with a size of about 175 μm, prepared by the plasma rotating electrode process (PREP). The chemical composition of the alloy powder is shown in Table 1. The alloy powder was dried at a temperature of 150 °C, holding for about 4 hours in a vacuum furnace. Before taking out the powder, the furnace was cooled to room temperature under a vacuum.

Fabrication of the LSFed Inconel 718 superalloy samples was carried out in a LDF-6000-60 laser solid forming system built by the Shenyang University of Aeronautics and Astronautics. The system was composed of six parts, including a 6550 W fiber laser generator (IPG Photonics Corporation, Oxford, MS, United States), an optical path transmission system, an inert gas processing room, a powder transmission system, an electromagnetic stirring system, and a numerical control table. The chemical composition of LSFed Inconel 718 superalloy is shown in Table 1. The LSF parameters used are listed in Table 2, and Ar gas was used for the powder feeding gas and the shielding gas. The size of the LSFed samples were 100 mm × 45 mm × 45 mm. During the forming process, the laser beam irradiated the surface of the substrate, and a molten pool formed on the substrate. At the same time, the metal powder was sent to the molten pool at a very high temperature through the powder feeding nozzle under the restriction of the powder gas. The powder melted quickly after entering the molten pool. With the movement of the laser beam, the rear side of the molten pool rapidly solidified on the surface of the relatively large substrate, and then the moving laser beam continuously created a new molten pool on the surface of the substrate in front of the original molten pool. The forming and solidification of the molten pool occurred continuously until the whole block was formed. In order to study the effect of electromagnetic field intensity on the precipitation of the δ phase, the electromagnetic field intensities were applied at different levels (0 mT, 30 mT, and 50 mT) in the process of LSF. The equipment for Laser Solid Forming (LSF) with electromagnetic stirring is shown in Figure 1.

To analyze the precipitation of the δ phase, the EMS-LSFed Inconel 718 superalloy samples (Shenyang University of Aeronautics and Astronautics, Shenyang, China) were cut into small blocks of about 8 mm × 8 mm × 2 mm by a wire electric discharge machine before heat treatment. The δ aging treatment was carried out in a SX2-5-12 box (Shanghai Xin Yi Instruments and Meters Company, Shanghai, China) resistance furnace at 950 °C for 1, 4, and 16 h and followed by water cooling to room temperature. The rated power of the SX2-5-12 box resistance furnace was 5 kW, the inside size of the furnace was 300 mm × 200 mm × 120 mm, and the rated temperature is 1200 °C.

To observe the morphology of the δ phase in the samples after different aging treatments, each small block was ground and polished with sandpaper and etched with a mixture of 50 ml HCl + 10 mL HNO_3_ + 2 mL HF + 38 mL H_2_O. Next, the microstructure was observed with a HITACHI S3400N scanning electron microscope (SEM, HITACHI, Shanghai, China). To quantitively analyze the amount and distribution of the δ phase, the Image Pro Plus image analysis software was used to get statistic data from SEM images. Meanwhile, the mechanical properties of the material were demonstrated through microhardness testing experiments. The microhardness of each aging samples was conducted on a WT-401MVD model hardness tester (Time-Top Company, Beijing, China). The load was 200 g and the holding time reached 10 s. Each sample was randomly measured for more than 30 points on the surface, and the average value was taken as the microhardness of the sample.

## 3. Results and Discussion

### 3.1. The “Cutting” and “Dissolution” of the Laves Phase Through the Precipitation of the δ Phase

Figure 2 shows the morphology and distribution of the δ phase after aging at 950 °C for different aging times from samples fabricated without electromagnetic stirring. It can be seen clearly that the size of the δ phase become larger and the volume fraction of the precipitation in interdendritic areas increases with an extension of aging time, as shown in Figure 2b–d. The precipitation of the δ phase is mainly concentrated around the Laves phase in interdendritic areas. Meanwhile, the δ phase nucleates around the Laves phase and grows in the form of short rods on both sides of the dendrite. This result is due to the fact that the nucleation and growth of the δ phase need the Nb concentration to remain above 6% molar fraction [29], and the Laves phase can provide enough Nb to precipitate the δ phase. Furthermore, there is an orientation relationship between needle-like δ phase and the Nb-rich Laves phase. This relationship is ${\left(010\right)}_{\delta }$//${\left(\bar{1}010\right)}_{Laves}$, ${\left[100\right]}_{\delta }$//${\left[\bar{1}2\bar{1}6\right]}_{L\mathrm{aves}}$ [30]. This explains the reason that the δ phase was precipitated in large Laves phase particles in the direct laser deposited (DLD) Inconel 718 superalloy.

The quantitative measurement of the δ phase is shown in Figure 3, and some results are listed in Table 3. It is observed that the average length of the δ phase increased alongside the prolonging of aging time, and the growth rate of the δ phase in the length direction between aging for 1 h and 4 h is faster than that from 4 h to 16 h in Figure 3a. Figure 3b shows that the average width of the δ phase increase slightly with aging time extended. As shown in Figure 3c, the length-width ratio of the δ phase was decreased slowly due to the fact that the growth rate of the δ phase in the length direction was larger than that in the width direction during aging from 1 h to 4 h. Meanwhile, the length–width ratio of the δ phase started to increase after aging for 4 h due to the fact that the growth rate of the δ phase in the width direction was larger than that in the length direction.

The nucleation of the δ phase needs a suitable position and sufficient alloying elements. Much more Nb and other alloying elements exist in the interdendritic areas, especially at the interface between the Laves phase and the γ (eutectic) phase, which satisfy the nucleation conditions, so the nucleation of the δ phase takes precedence around the Laves phase. The growth of the δ phase needs a supply of the Nb element, not only from Laves phase, but also from the γ (eutectic) area and the γ (matrix) in the core dendrite with the δ aging time extended, as shown in Table 4. The γ (matrix) phase (or γ-phase) is the austenite matrix phase, and its crystal structure is a face-centered cubic structure. The γ (eutectic) phase is the Laves phase + the γ (matrix) phase, which is called the eutectic phase. Comparing Figure 4e,f, the large content of Nb in the γ (eutectic) area decreases sharply from 9.1% to, 6.3% which promotes the δ phase to grow in the length direction from the Laves phase to the γ (eutectic) area. However, the δ phase stops growing in the length direction when it grows close to the core dendrite and does not form the long needle-like shaped δ phase to run through the grain, because the content of Nb is only 3.5%, which cannot promote the δ phase to grow into the core dendrite in the length direction, as shown in Figure 4b,d,f and Figure 5. Then, the δ phase begins to grow along the width direction when the aging time continues to be prolonged because the concentration of Nb in front of the δ phase tip close to the core dendrite is lower than that on the side of the δ phase in the γ (eutectic) area, and the driving force of the lateral growth of the δ phase is greater than that of the continuous growth in the core dendrite along the length direction. Therefore, the growth rate of the δ phase along the width direction is faster than that along the length direction at this time, which causes the length–width ratio of the δ phase to being to increase.

It can also be seen that the precipitate and growth of the δ phase consume Nb in the Laves phase with the δ aging time increased (compare Figure 4a,c,e with Figure 4b,d,f. The content of Nb decreases from 25.3% to 24.8% in the Laves phase after the aging time increases (Table 4), which promotes the δ phase to precipitate and grow. This also means that the precipitation and growth of the δ phase accompanied by the “cutting” and “dissolution” of the Laves phase, and this process can lead to a significant change in morphology and a decrease in the volume fraction of the Laves phase compared to the long-strip shaped Laves phase in Figure 2a.

### 3.2. Effect of Electromagnetic Filed Intensity on Precipitation of the δ Phase

Figure 6 and Figure 7 show the morphology and distribution of the δ phase in the EMS-LSFed Inconel 718 superalloy samples with different electromagnetic field intensities after aging at 950 °C for different aging times. It was clearly found that the morphology of the δ phase converts from a rod-like shape to a long-needle shape, and these δ phases tend to grow in a direction from the inter dendrite to the core dendrite, with the electromagnetic field intensity increasing (compare Figure 2d, Figure 6d, and Figure 7d). Further, the morphology of the Laves phase changes from a long-strip to a granular shape with the growth of the δ phase in samples influenced by the high electromagnetic field intensity in Figure 7a–d. The changes of the δ phase’s morphology and distribution can be explained by the redistribution of the alloying elements, resulting from the convection of liquid metal in the molten pool during EMS-LSF. With a longer aging time, the length of the δ phase increases correspondingly, as shown in Figure 7b–d, which agrees with Figure 2. When the aging time is increased to 16 h, no significant changes are found in the morphology and amount of the δ phase in samples fabricated with a low electromagnetic field intensity compared to Figure 2d and Figure 6d. However, significant changes in length are found in samples fabricated with a higher electromagnetic field intensity (Figure 7d). It can be seen clearly that the long needle-like δ phases are formed, and they grow through all the inner dendrites when the electromagnetic field intensity reaches 50 mT.

Electromagnetic stirring with the Lorentz force can lead to the convection of liquid metal in a molten pool and result in the rapid redistribution of alloying elements in front of the solid/liquid interface [21]. The enrichment of the alloying elements in front of the solid/liquid interface was beneficial to the formation of the eutectic Laves + γ phase. Serious convection caused by electromagnetic stirring produced plenty of alloying elements, which enriched in front of the solid/liquid interface far into the liquid metal. Meanwhile, this process of convection reduced the enrichment of the alloying elements in the inter dendrite and consequently eliminated the formation of the Laves + γ eutectic reaction [31]. Therefore, electromagnetic stirring can relieve segregation during solidification, which means that the difference in the content of Nb between the inter dendrites and core dendrites can be reduced.

Quantitative measurement of the δ phase in the EMS-LSFed Inconel 718 superalloy with different electromagnetic field intensities (Figure 8a) established that the average length of the δ phase had an increasing trend with the addition of electromagnetic field intensity at different aging times. To be specific, the average length of the δ phase grew faster than that its average width when the electromagnetic field intensity increased from 0 mT to 50 mT in Table 5 and Table 6. Therefore, there is a slow decline in the length-width ratio of the δ phase, which was attributed to the fact that the extent of the length direction is much larger than that of the width direction (Table 7).

This phenomenon also indicated that the absorption of Nb in the front region of the δ phase led to growth in the length direction during the growth of the δ phase [30]. There was not enough Nb in the core dendrite, which caused the phenomenon that the δ phase could not grow continuously into the core dendrite without electromagnetic stirring. By contrast, the electromagnetic field produced a strongly stirring influence on the solid/liquid interface in the molten pool, which reduced the enrichment of the alloying elements in the inter dendrite [32]. Meanwhile, the content of Nb in the core dendrite increased from 2.4% to 6.5% with an increase of the electromagnetic field intensity, which could promote the growth and precipitation of the δ phase from the inter dendrite to the core dendrite, as shown in Table 8. Therefore, based on the combined reference and test result, it can be concluded that the electromagnetic stirring gradually homogenizes the distribution of the alloying elements. Due to the increase of the strengthening element Nb in the core dendrite, the δ phase gradually grows and extends to cross the core dendrite, as shown in Figure 7d.

The volume fraction of the δ phase was measured quantitively by measuring the area fraction of the δ phase in SEM images and then calculating the volume fraction of the δ phase according to the stereological interchange formula: V_V_ = A_A_ = L_L_ = P_P_ [33]. A_A_ is the area fraction, which means that a phase takes up a fraction of the total area of the sample in the area of the picture. V_V_, L_L_, P_P_ are the volume fraction, line fraction, and point fraction. Figure 9 shows the quantitative results of the volume fraction of the δ phase in the EMS-LSFed Inconel 718 superalloy samples prepared with different electromagnetic field intensities after aging at 950 °C for different times. Overall, the volume fraction of the δ phase monotonically increases with the aging time and electromagnetic field intensity increasing (Table 9). Moreover, increasing the electromagnetic field intensity can promote the precipitation of the δ phase, and the growth rate of the δ phase increases accordingly. In other words, when the electromagnetic field intensity stayed at 0 mT, the δ phase increased rapidly between 4 h and 16 h, as shown in Figure 9. Then, the rapid growth of the δ phase advanced between 1 h and 4 h, and the volume fraction of the δ phase sharply reached up to 11.89 vol.% when the electromagnetic field intensity reached 30 mT, and the volume fraction of the δ phase increased slowly after aging 4 h. Then, when the electromagnetic field intensity increased to 50 mT, the volume fraction of the δ phase after aging 1 h become quite large at 16.18 vol.% and did not increase much as the aging time increased, thereby illustrating that the period with the largest increase in the volume fraction of the δ phase would occur within 1 h. It was indicated that the precipitation time for large amounts of the δ phase was clearly shortened when the electromagnetic field intensity increased.

In general, the growth of the δ phase requires a continuous supply of the Nb element, and electromagnetic stirring can make the distribution of the Nb element more uniform in the dendrite, thereby increasing the content of the Nb element in the core dendrite. This will be beneficial to the growth of the δ phase in the length direction, so then the δ phase grows into the core dendrite to enhance the homogeneity of the microstructure.

### 3.3. Precipitation Kinetics Analysis of Phase Precipitation

The relationship between the volume fraction of δ phase and the aging time can be described by the Avrami equation [34,35,36]:
(1)$$V_{\delta }=V_{S,\delta }\left[1-\exp\left(-\alpha \cdot t^{n}\right)\right]$$
where *V_δ_* is the volume fraction of the δ phase, *V_s,δ_* is the saturated value of δ phase precipitation, n is time index, and α is precipitation rate of the δ phase.

According to the experimental results in Figure 9, the relationship between ln[−ln (1 − *V_δ_/V_S,δ_*)] and ln*t* was fit, as shown in Figure 10. Then, the precipitation kinetic parameters of the δ phase were obtained through a regression analysis, and the results are listed in Table 10. It is seen that the value of the precipitation rate α increased with the intensity of the electromagnetic field rising, and the maximum increase occurred between 30 mT and 50 mT, which rapidly increased from 0.07135 μm/s to 0.56738 μm/s. The time index n is the slope of the straight line in Figure 10. The time index n gradually decreased with an increase of the electromagnetic field intensity, and it decreased by about half when the electromagnetic field intensity increased from 30 mT to 50 mT. When time index n is smaller with the electromagnetic field intensity increasing, the slope of the straight line is also lower, which means that the ordinate at the origin (the precipitation rate of the δ phase α) becomes larger. Therefore, a large amount of the δ phase can precipitate in a shorter time if the electromagnetic field intensity is too high.

Nb is an important element when forming the δ phase. The convection process intensifies in the molten pool, with an increase in the electromagnetic field intensity, resulting in more uniform distribution of Nb, which can lead the δ phase to grow from the inter dendrite to the core dendrite after aging treatment. In addition, the precipitation rate of the δ phase becomes larger when the electromagnetic field intensity continues to increase. This occurs because an intensification of the convection process causes more Nb to be distributed in the core dendrite after electromagnetic stirring. According to the constitutional supercooling criterion, a high solute concentration can promote the nucleation and growth of phase rapidly. The content of Nb increases in the core dendrite after electromagnetic stirring, which can satisfy the conditions of nucleation and growth for the δ phase. Therefore, a large amount of the δ phase precipitates rapidly from the inter dendrite to the core dendrite with a high electromagnetic field intensity.

### 3.4. Electrochemical Corrosion of the Weld

The microhardness of the EMS-LSFed Inconel 718 superalloy samples varied with different aging times, as shown in Figure 11. It was clearly found that the microhardness falls with the aging time prolonged (Table 11). The microhandness decreases because the hard Laves phase has dissolved gradually with the aging time prolonged. Meanwhile, the nucleation and growth of the δ phase need to consume plenty of Nb in the Laves phase and will dissolve the Laves phase and eliminate the segregation of the Nb element in the inter dendrite. Furthermore, the precipitation of the δ phase only occurs in the interdendritic areas with short rod-like shapes. The precipitation also indicates that the Nb element only diffused into the interdendritic areas and cannot be diffused into the core dendrite to strengthen the matrix’s γ phase during aging treatment.

However, the microhardness of the EMS-LSFed Inconel 718 superalloy samples increased gradually with an increase in the electromagnetic field intensity, as shown in Figure 11. It can be seen that the microhardness of the samples increased, with the electromagnetic field intensity extending in different aging times, as shown in Table 11. In addition, compared to the original sample, for which the microhardness reached 300 HV without the electromagnetic field intensity and heat treatments, the volume fraction of the δ phase in the aging samples was larger than 12%, and the microhardness exceeded 300 HV with the effect of electromagnetic field intensity and heat treatment. It can be seen that electromagnetic stirring slightly increased the microhardness of the samples because it can sharply promote the distribution of Nb in the γ-phase matrix and largely enhance the precipitation of the long needle-like δ phase from the inter dendrite to the core dendrite.

The precipitation of the δ phase first occurs in the interdendritic areas with short rod-like shapes and consumes Nb in the areas of the Laves phase and the γ (eutectic) phase, so that Nb cannot diffuse from the interdendritic areas into the core dendrite to strengthen the matrix after heat treatment. After that, the δ phases grow out of the inter dendrite with a long needle-like shape and absorbs the Nb element from the γ-phase matrix in the core dendrite (Figure 12). As mentioned above, the content of Nb in the core dendrite is higher when electromagnetic stirring is applied. Therefore, a greater quantity of the δ phase with a long needle-like shape uniformly precipitating from the inter dendrite to the core dendrite strengthens the matrix and obviously improves the microhardness.

## 4. Conclusions

Electromagnetic stirring assisted laser solid forming in an Inconel 718 superalloy makes the distribution of the alloying elements (especially Nb) more homogeneous. Since Nb is the main formation element of the δ phase, the method of EMS-LSF can change the morphology and distribution of the δ phase by controlling the distribution of Nb. It can be found that the morphology of the δ phase converts from rod-like to a long-needle shape, and its distribution tends to change from the inter dendrite to the core dendrite by increasing the electromagnetic field intensity. Meanwhile, the precipitation of the δ phase is accompanied by the “cutting” and “dissolution” of the Laves phase during the δ aging treatment. This result is due to the fact that the growth of the δ phase needs a continuous supply of the Nb element from the adjacent Laves phase. The precipitation rate of the δ phase increased when the electromagnetic field intensity increased. This increase is attributable to the higher Nb content in the core dendrite, which was caused by the serious convection of liquid metal. Finally, electromagnetic stirring can slightly improve the microhardness of the samples due to the higher Nb content in the core dendrite, and the more homogeneous distribution of the solute elements resulted from the serious convection of the liquid metal, which can, therefore, strengthen the matrix.

## Figures and Tables

**Figure 1 materials-12-02604-f001:**
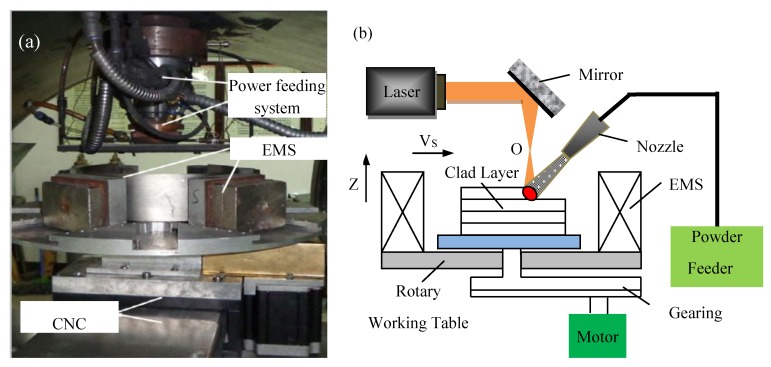
The equipment for laser solid forming (LSF) with electromagnetic stirring: (**a**) picture; (**b**) schematic diagram.

**Figure 2 materials-12-02604-f002:**
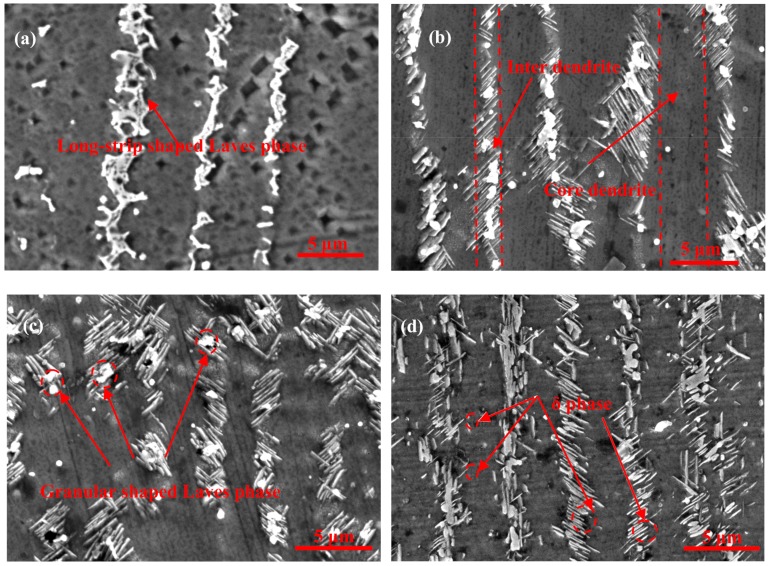
The morphology and distribution of the δ phase after aging at 950 °C for different aging times for samples fabricated without electromagnetic stirring: (**a**) the as-deposited sample, (**b**) aging for 1 h, (**c**) aging for 4 h, and (**d**) aging for 16 h.

**Figure 3 materials-12-02604-f003:**
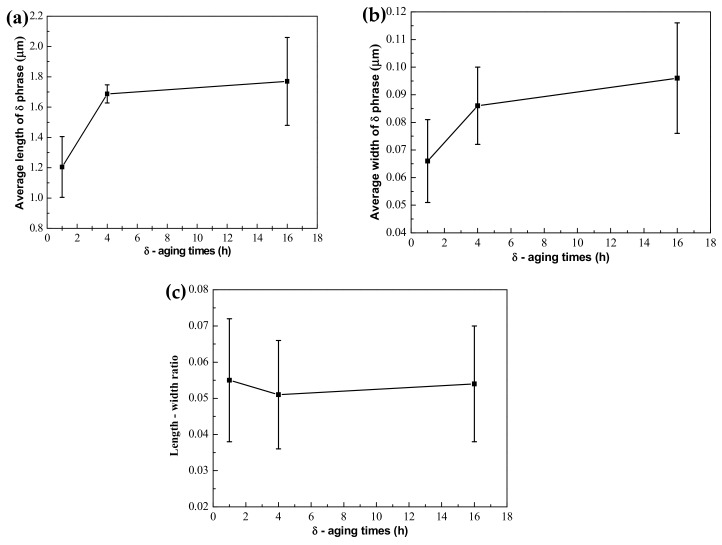
Quantitative measurements of the δ phase in the LSFed Inconel 718 superalloy after ageing treatment ((**a**–**c**) are average length, average width, and length–width ratio of the δ phase, respectively).

**Figure 4 materials-12-02604-f004:**
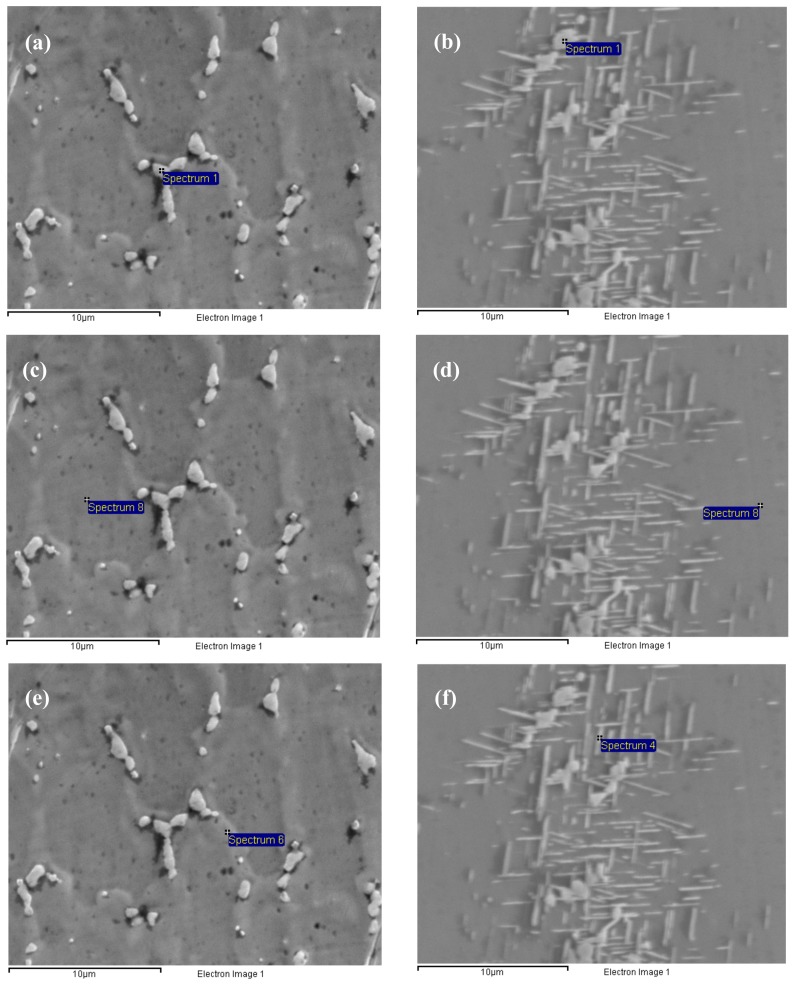
Energy dispersive spectrometer measurement for the Inconel 718 with the δ aging for 0 h in (**a**,**c**,**e**), and with aging for 12 h in (**b**,**d**,**f**). The locations of the measurement are in the Laves phase (**a**,**b**); the core dendrite (**c**,**d**); the γ (eutectic) area (**e**,**f**).

**Figure 5 materials-12-02604-f005:**
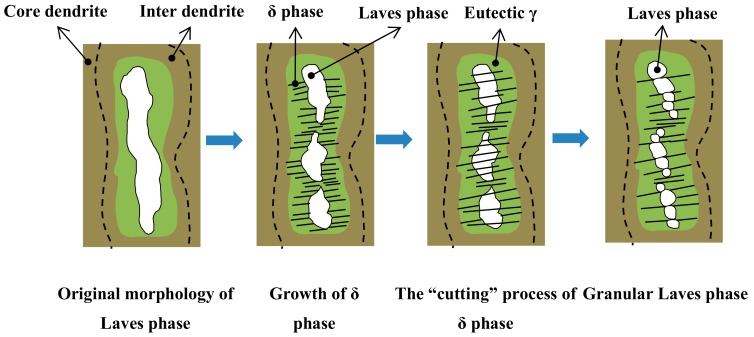
Schematic diagram of the “cutting” and “dissolution” of the Laves phase through the precipitation of the δ phase.

**Figure 6 materials-12-02604-f006:**
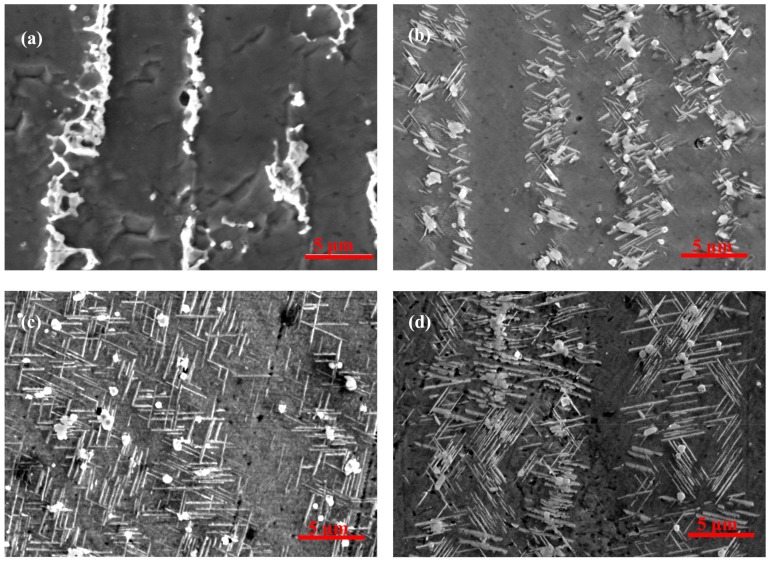
Morphology and distribution of the δ phase after aging at 950 °C for different aging times for samples fabricated with electromagnetic stirring at 30 mT. (**a**) The as-deposited sample with 30 mT, (**b**) aging for 1 h, (**c**) aging for 4 h, and (**d**) aging for 16 h.

**Figure 7 materials-12-02604-f007:**
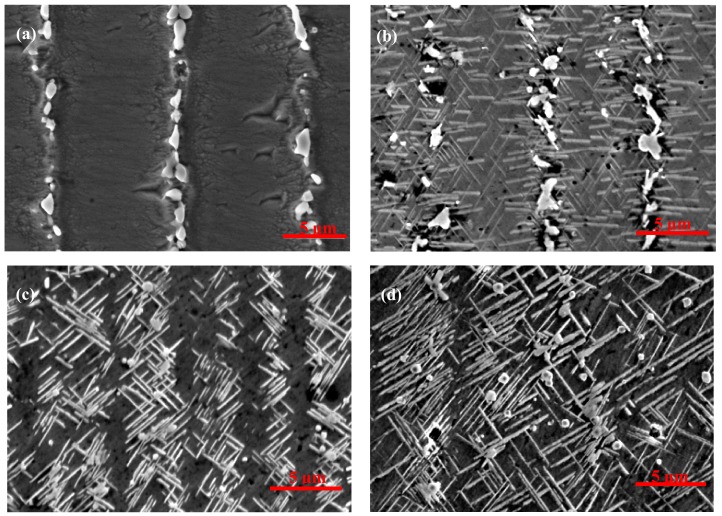
Morphology and distribution of the δ phase after aging at 950 °C for different aging times for samples fabricated with electromagnetic stirring at 50 mT. (**a**) The as-deposited sample with 50 mT, (**b**) aging for 1 h, (**c**) aging for 4 h, and (**d**) aging for 16 h.

**Figure 8 materials-12-02604-f008:**
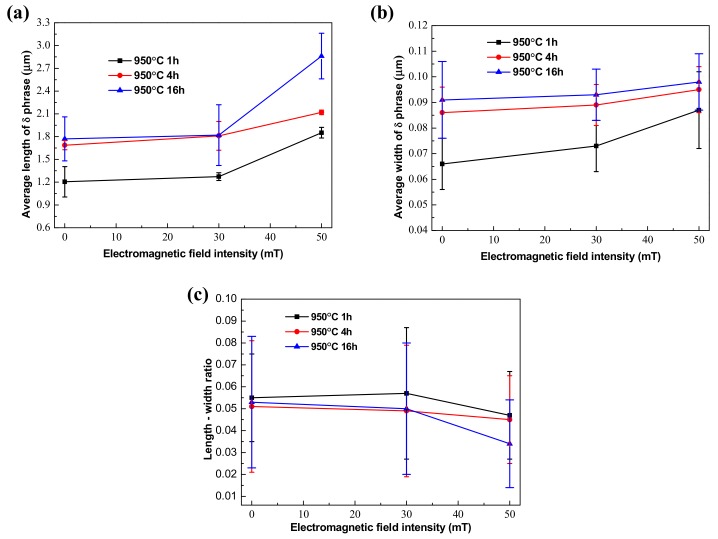
Quantitative measurements of the δ phase in EMS-LSFed Inconel 718 superalloy samples fabricated with different electromagnetic field intensities. (**a**–**c**) are average length, average width, and length-width ratio of the δ phase, respectively).

**Figure 9 materials-12-02604-f009:**
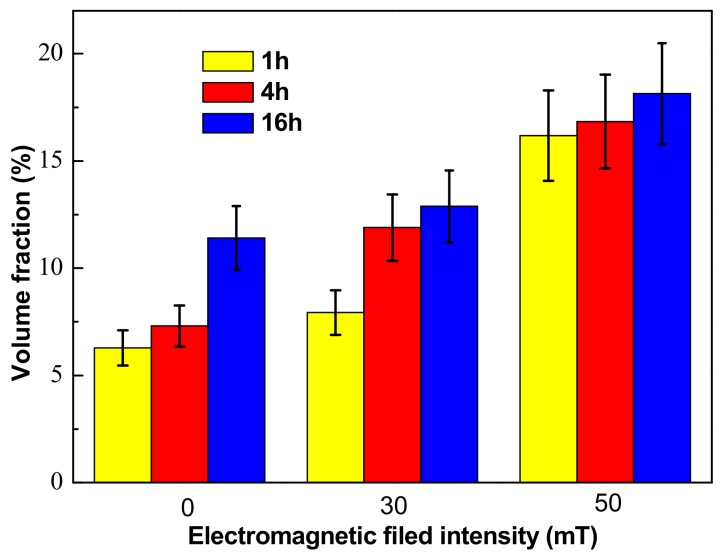
Volume fraction of the δ phase in the EMS-LSFed Inconel 718 superalloy samples prepared with different electromagnetic field intensities.

**Figure 10 materials-12-02604-f010:**
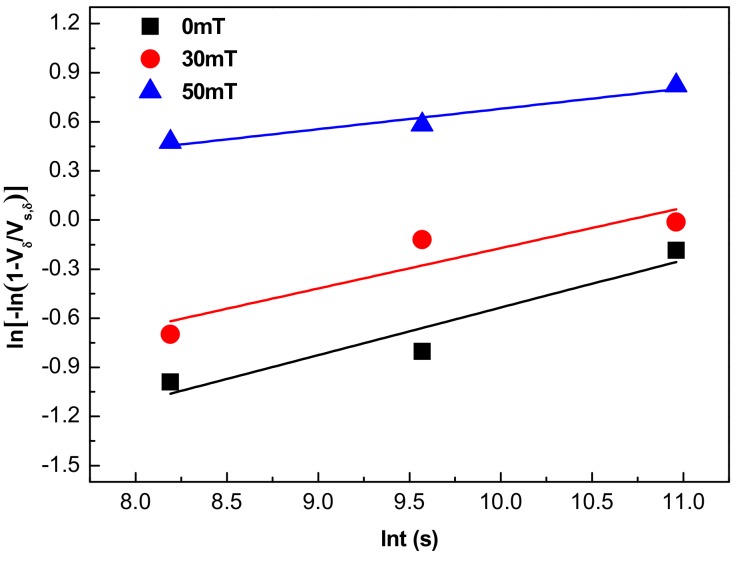
The relationship between ln[−ln (1 − *V_δ_/V_S,δ_*)] and ln*t.*

**Figure 11 materials-12-02604-f011:**
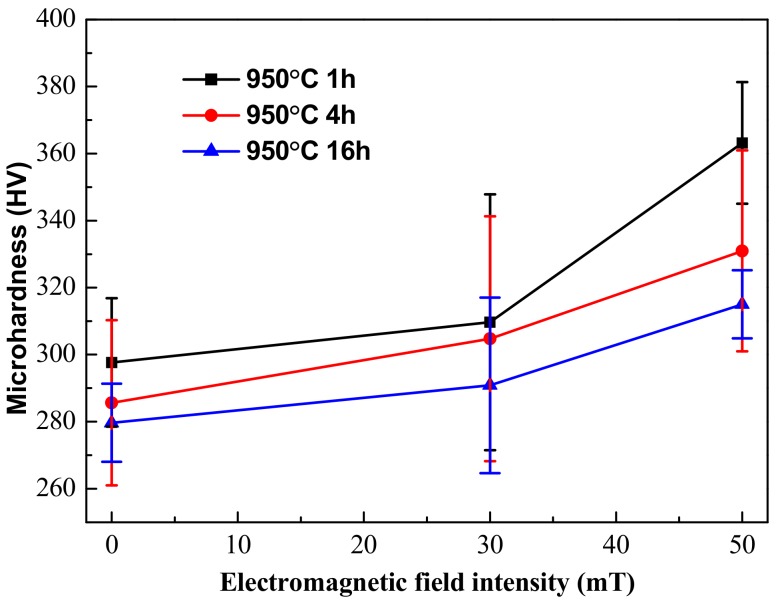
Microhardness of the δ phase in the EMS-LSFed Inconel 718 superalloy samples fabricated with different electromagnetic field intensities: (**a**) 0 mT, (**b**) 30 mT, and (**c**) 50 mT.

**Figure 12 materials-12-02604-f012:**
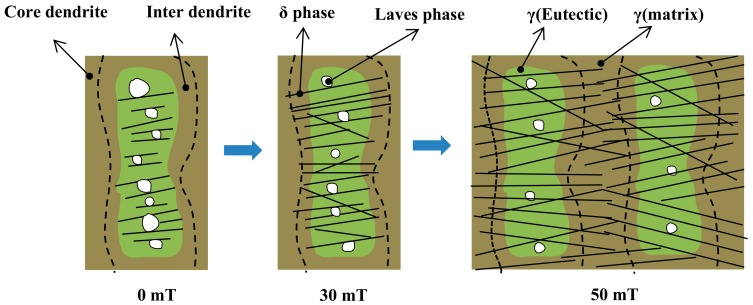
Schematic diagram of the effect of micromagnetic field intensity on the morphology and distribution of the δ phase.

**Table 1 materials-12-02604-t001:** Chemical composition of the Inconel 718 superalloy powder and the laser solid forming (LSF) sample (wt.%).

Material	C	Nb	Cr	Fe	Al	Ti	Mo	Ni
Powder	0.03	5.2	19.7	18.4	0.6	1.0	3.0	Bal.
LSF sample	0.05	5.6	19.6	18.2	0.5	1.1	3.3	Bal.
AMS: 5663	0.08 max	4.75–5.5	17–21	16–20	0.2–0.8	0.65–1.15	2.8–3.3	Bal.

Note: the content of C is analyzed by optical emission spectrometer, and the others are obtained by energy dispersive spectrometer (average content of 6 different points). AMS means aerospace material specification.

**Table 2 materials-12-02604-t002:** Laser solid forming parameters of the Inconel 718 superalloy.

Laser Power (W)	Scanning Speed (mm/s)	Powder Supply Rate (g/min)	Shielding Gas Flux (L/min)	Spot Diameter (mm)	Overlap of Adjacent Passes (%)	Δ Z (mm)
2000–2200	6–8	8–12	6	3	25–40	0.3

**Table 3 materials-12-02604-t003:** The relation between the size of the δ phase and the aging time (unit: μm).

Aging Time	1 h	4 h	16 h
Length	1.205	1.687	1.770
Width	0.066	0.086	0.096
Length-width ratio	0.055	0.051	0.054

**Table 4 materials-12-02604-t004:** EDS measurement of Nb in different locations (wt.%).

	Laves Phase	γ (Eutectic)	Core Dendrite
Aging 0 h	25.3	9.1	2.5
Aging 12 h	24.8	6.3	3.5

**Table 5 materials-12-02604-t005:** The influence of electromagnetic field on the average length of the δ phase after different aging times.

Aging Time	0 mT	30 mT	50 mT
1 h	1.205 μm	1.273 μm	1.852 μm
4 h	1.687 μm	1.810 μm	2.120 μm
16 h	1.770 μm	1.820 μm	2.861 μm

**Table 6 materials-12-02604-t006:** The influence of the electromagnetic field within the average width of the δ phase for different aging times.

Aging Time	0 mT	30 mT	50 mT
1 h	0.066 μm	0.073 μm	0.087 μm
4 h	0.086 μm	0.089 μm	0.095 μm
16 h	0.091 μm	0.093 μm	0.098 μm

**Table 7 materials-12-02604-t007:** The influence of the electromagnetic field within the length–width ratio of δ phase for different aging times.

Length–Width Ratio	0 mT	30 mT	50 mT
1 h	0.055	0.057	0.047
4 h	0.051	0.049	0.045
16 h	0.053	0.050	0.034

**Table 8 materials-12-02604-t008:** EDS measurement of Nb in the core dendrite of the samples fabricated with different electromagnetic field intensities (wt.%).

	0 mT	30 mT	50 mT
Core dendrite	2.4	5.8	6.5

**Table 9 materials-12-02604-t009:** The influence of the electromagnetic field within the average volume fraction of the δ phase for different aging times (vol.%).

	0 mT	30 mT	50 mT
1 h	6.284	7.937	16.183
4 h	7.316	11.895	16.841
16 h	11.413	12.880	18.142

**Table 10 materials-12-02604-t010:** Kinetic parameters of the δ phase precipitation in the EMS-LSFed Inconel 718 superalloy samples prepared with different electromagnetic field intensities.

Electromagnetic Field Intensities/mT	Precipitation Rate *α*	Time Index *n*
0	0.032	0.290
30	0.071	0.247
50	0.567	0.125

**Table 11 materials-12-02604-t011:** The influence of the electromagnetic field on the microhardness of samples with different aging times.

Microhardness	0 mT	30 mT	50 mT
1 h	297.70	309.69	363.16
4 h	285.66	304.80	330.95
16 h	279.69	290.86	315.02

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
