# Peer review of "The δ Phase Precipitation of an Inconel 718 Superalloy Fabricated by Electromagnetic Stirring Assisted Laser Solid Forming"

_materials, 2019, doi:10.3390/ma12162604_

Round 1
Reviewer 1 Report
This paper introduces an interesting difference in morphology and distribution of δ phase in Inconel 718 according to the applied electromagnetic intensity during laser solid forming (LSF). By applying electromagnetic field of 30 or 50 mT to IN718 during depositing, the dendrite formation and distribution can be clearly changed and the microhardness is also enhanced. I, one of the reviewers, consider that metal structure formation is an important issue for metal additive manufacturing (AM) because many researchers have already found that the mechanical characteristics in AM parts are difficult to ensure without any heat treatments. In most metal AMs, dendrite growth is not avoidable and how to deal with it is a significant point to make the quality of AM part high and stable. Against this background, this paper provides benefit results for controlability of metal structure from the viewpoint of electromagnetic action which is hardly discussed by other studies. It should be shared among researches focusing on metal AM.
In order to improve the quality of this paper, I would like to provide several comments below. Please discuss on my comments and revise your paper.
General comment
* I found many mistakes in English expression in your paper. For example, in 37th line of 1st page, “casting and forging process” should be “casting and forging processes” and “LSF bring” should be “LSF brings.” In 39th line of 1st page, “LSFed Inconel 718 parts has” needs to be changed to “LSFed Inconel 718 parts have.” Because these small mistakes were found many times, I recommend you to receive a native check on your paper.
Abstract
It is written well.
1. Introduction
It introduces relative works well. The purpose of this study is also clearly shown.
2. Experimental methods
* You should show a picture of LSF used in your study. Considering the journal “Materials” is read by various researchers in many fields, many of them would not be familiar with AM process. In addition, the reader may want to know how you setup an electromagnetic system into LSF machine.
* More information for “SX3-14-10 box” is needed. Its properties also influence on the experimental results.
3. Results and discussion
* The direction of electromagnetic field also would be important information for your study. I am wondering whether you can find any relationships between the directions of dendrite growth and electromagnetic field. Focusing on Figs. 5(c) and 6(b), the most δ phase grew in the horizontal direction, although the δ phase hardly grew in the horizontal or vertical directions in Figs. 5(d), 6(c) and 6(d). Would you add a discussion into your paper, why the direction of δ phase growth changes in each experiment?
* In Figs. 1, 5 and 6, the color of letters and scale bars should be changed. It is hard to understand if you use white letters on a gray-scale picture.
* In Figs. 1(b), 4 and 11, the direction of arrows would be upset. The arrow should point at what you want to emphasize.
4. Conclusion
It summarizes the content of this paper correctly.
Author Response
1. General comment
* I found many mistakes in English expression in your paper. For example, in 37th line of 1stpage, “casting and forging process” should be “casting and forging processes” and “LSF bring” should be “LSF brings.” In 39th line of 1st page, “LSFed Inconel 718 parts has” needs to be changed to “LSFed Inconel 718 parts have.” Because these small mistakes were found many times, I recommend you to receive a native check on your paper.
Thank you for your comment.
We have already corrected some mistakes in grammar in the article, such as the following,
(1) In 35th line of 1stpage, “casting and forging process” has been corrected to “casting and forging processes”.
(2) In 35th line of 1stpage, “LSF bring” has been corrected to “LSF brings”.
(3) In 39th line of 1st page, “LSFed Inconel 718 parts has” has been corrected to “LSFed Inconel 718 parts have.”
2. Experimental methods
* You should show a picture of LSF used in your study. Considering the journal “Materials” is read by various researchers in many fields, many of them would not be familiar with AM process. In addition, the reader may want to know how you setup an electromagnetic system into LSF machine.
* More information for “SX3-14-10 box” is needed. Its properties also influence on the experimental results.
Thank you for your comment.
(1) We have already added some pictures about EMS + LSF technology in the article.
(2) We get the wrong type of the box resistance furnace, and “SX3-14-10 box resistance furnace” should be corrected to “SX2-5-12 box resistance furnace” in the article. We have already given more details about SX2-5-12 box resistance furnace, for example, the rated power of it is 5 KW, then the inside size of furnace is 300 mm× 200 mm× 120 mm, and the rated temperature is 1200 ℃.
3. Results and discussion
* The direction of electromagnetic field also would be important information for your study. I am wondering whether you can find any relationships between the directions of dendrite growth and electromagnetic field. Focusing on Figs. 5(c) and 6(b), the most δ phase grew in the horizontal direction, although the δ phase hardly grew in the horizontal or vertical directions in Figs. 5(d), 6(c) and 6(d). Would you add a discussion into your paper, why the direction of δ phase growth changes in each experiment?
* In Figs. 1, 5 and 6, the color of letters and scale bars should be changed. It is hard to understand if you use white letters on a gray-scale picture.
* In Figs. 1(b), 4 and 11, the direction of arrows would be upset. The arrow should point at what you want to emphasize.
Thank you for your comment.
(1) Electromagnetic field can not directly affect the growth direction of δ phase, but it can bring an significant influence in the growth rate of δ phase. As we all known, the crystal orientation relationship between δ phase and matrix γ phase is //、// , which means that the direction of δ phase depends on the direction of γ phase. In addition, there is an orientation relationship between δ phase and Laves phase, which is //、//. Therefore, the direction of δ phase growth changes is attributed to the electromagnetic field effect on γ phase and Laves phase.
(2) In Figs. 1, 5 and 6, we have changed the color of letters and scale bars from white to red on a gray-scale picture.
(3) In Figs. 1(b), we have changed the direction of arrows, and the direction indicated by the arrow is the area of text description. In Figs. 4 and 11, we have added spots at the end of the arrows, which means that the area where the dot is located is the area of text description.

Reviewer 2 Report
It can be published in the current form
Author Response
Thanks the reviewer for you time and patient on the manuscript.
Reviewer 3 Report
The introduction is not written in detail.
The scatter of measurements in Fig 2 and 7 is too large and does not allow to draw any conclusions and arguments about changing of the Average length
Few research methods were used in the work. It would be useful to present the results of the X-ray study.
Author Response
The introduction is not written in detail.
The scatter of measurements in Fig 2 and 7 is too large and does not allow to draw any conclusions and arguments about changing of the Average length
Few research methods were used in the work. It would be useful to present the results of the X-ray study.
Thank you for your comment.
(1) We believe we have already give more details in introduction. For example, the first three paragraphs give the reason why we choose the material like Inconel 718 superalloy and why we use LSF technology to manufacture samples. The 4th paragraph tell us why we cannot use traditional high temperature solid solution treatment and the 5th and 6th paragraph let us know why we should combine EMS technology with LSF technology. In addition, we cite plenty of references to prove it. The last paragraph summed up the purpose and the significance in this paper. If we add other contents in introduction, Introduction section is quite long for the main contents, which should be summarize briefly.
(2) In Fig 1, 5 and 6, it can be found clearly that there is long and short δ phase in SEM picture. Therefore, we can get a relatively large range in Fig 2 and 7 with quantitative statistics. For the same reason, quantitative statistics with a relatively large range in width is similar as in length, meanwhile, the figure is account to two decimal places for the width of δ phase, which means that the change of the width is very small that is still keep two decimal places in Fig 2 and 7.
(3) Our paper applies the method of quantitative statistics and qualitative analysis to the research the precipitation rule of δ phase. We believe it is enough to use these research methods. Qualitative analysis with SEM picture can get change rule of δ phase entirety. Quantitative statistics not only can make us understand change range and change trend of δ phase in line picture, but also can give EDS results which is beneficial to know the process of precipitated δ phase under EMS specifically.
Reviewer 4 Report
The authors present work is documenting a microstructural change caused by applying electromagnetic field in Inconel 718 superalloy. This result is significant and interesting. But, before publishing, the authors should reconsider about following points.
1. In p2 line66, please replace °F with °C.
2. In p2 line58, Does “our material” mean Inconel 718 superalloy?
3. Introduction section is quite long for the main contents. I recommend to reconsider and summarize briefly.
4. What are δ, γ and γ” phase? It is better to add the compositions or crystal structures of those phase when the authors use them first.
5. In p3 line 93, a space should be added between 175 and μm.
6. In the experimental section, the authors mention that the molten pool formed on the substrate when the laser beam irradiated. I’m concerned that the malted substrate and metal powder would be mixed and the composition of superalloy would change. Did the authors investigate the composition of the superalloy after laser solid forming process? The composition should be clearly stated in the paper. I’m concerned that the composition of the samples and also δ would change by electromagnetic stirring.
7. In Fig. 1(b), there is no scale bar.
8. In Fig. 1(a), I can find numerous voids or hole while there are few voids in Fig. 1(b). Could the authors explain why did the voids disappear after aging treatment?
9. In p4 line 144, what is IN718? Is that same with Inconel 718 superalloy?
10. I suppose that the unit of horizontal axis of Fig. 2 is “h”.
11. Please add the unit in Table 3.
12. In p6, the authors mentioned that the Ni content of γ decreased during aging and then δ precipitated. However, I suppose that the opposite phenomenon happened: δ precipitated and then the Ni content of γ decreased.
13. The authors used EDS to measure the Nb content. However, I recommend to use WDS by using EPMA because EDS is not suitable for a quantitative analysis. I recommend to investigate the composition of δ phase in addition to other phases.
14. Why did the authors use the sample aged for 12 h for EDS? Both Fig. 1 (a) and Fig. 3(a) is obtained from the sample before aging, but the microstructure looks quite different. Why? In addition, Fig. 3(a), (c) and (e) are images observed from the same region. So, it is better to show measurement point in one image. In Fig. 3(f), the measurement location looks indicate δ phase, not γ phase.
15. How did the authors judge that the shape of δ phase changed from rod-like to needle-like?
16. p8 line 208-210, the authors mention that there is no significant difference between Fig. 1(d) and Fig. 5(d). However, the length, the number of precipitated δ and the number of preferential growth direction of δ in Fig. 5(d) is larger than those in Fig. 1(d). In addition, I suppose that the increase of the number of preferential growth direction of δ phase is one of the big effects of the addition of electromagnetic field while the authors don’t mention about that.
17. In Fig. 10, the error bars are quite big. I suppose that the load was too small and the authors measured hardness of a limited region. I recommend to increase the load and measure the hardness from larger region.
18. What are VV, AA, LL and PP?
19. For fitting analysis like Fig. 9, it is better to use over four points.
20. Why did the time index n change drastically when electromagnetic field intensity was set to 50 mT? Please explain the relationship with the precipitation kinetics.
21. Line 326-328 is only about the sample with electromagnetic field of 0 mT?
22. How much is the hardness before aging treatment?
Lastly, there are numerous mistakes and errors in sentence structure in the manuscript. Please consider proof reading either using a proof-reading service or a native speaker with a scientific background.
I suppose that “believe” and “someone” are not used in a scientific paper.

Author Response
Thank you for your comment.
1. In p2 line66, please replace °F with °C.
We have already replaced °F with °C in p2 line 64.
2. In p2 line58, Does “our material” mean Inconel 718 superalloy?
In p2 line56, “our material” has been corrected to “the material”.
3. Introduction section is quite long for the main contents. I recommend to reconsider and summarize briefly.
We believe we have already give more details in introduction which is not too long. For example, the first three paragraphs give the reason why we choose the material like Inconel 718 superalloy and why we use LSF technology to manufacture samples. The 4th paragraph tell us why we cannot use traditional high temperature solid solution treatment and the 5th and 6th paragraph let us know why we should combine EMS technology with LSF technology. In addition, we cite plenty of references to prove it. The last paragraph summed up the purpose and the significance in this paper. If we delete and short some contents in introduction, Introduction section is incomplete and incomprehensible for the main contents, which should be give more details in introduction.
4. What are δ, γ and γ” phase? It is better to add the compositions or crystal structures of those phase when the authors use them first.
We have already add the compositions or crystal structures of δ, γ and γ” phase in our paper.
δ phase is a stable phase in an orthorhombic system, and its molecular formula is Ni3(Nb0.8Ti0.2).
γ phase is austenite matrix phase and its crystal structure is face-centered cubic structure.
γ” phase is the metastable phase of the body-centered tetragonal system, which belongs to Ni3Nb phase.
5. In p3 line 93, a space should be added between 175 and μm.
In p3 line 93, we have already added space between 175 and μm.
6. In the experimental section, the authors mention that the molten pool formed on the substrate when the laser beam irradiated. I’m concerned that the malted substrate and metal powder would be mixed and the composition of superalloy would change. Did the authors investigate the composition of the superalloy after laser solid forming process? The composition should be clearly stated in the paper. I’m concerned that the composition of the samples and also δ would change by electromagnetic stirring.
As we all known, there are very low and more local heat input for laser solid forming technology. Therefore, laser solid forming process can cause very low dilution rate of matrix materials. In addition, we choose the middle section of the laser solid forming samples as an experimental sample, which means that the composition of superalloy would not change in the experiment.
7. In Fig. 1(b), there is no scale bar.
We have already add scale bar in In Fig. 1(b).
8. In Fig. 1(a), I can find numerous voids or hole while there are few voids in Fig. 1(b). Could the
authors explain why did the voids disappear after aging treatment?
We can find numerous voids or hole in Fig. 1(a) because overweight corrosion of these samples results in pitting corrosion. However, corrosion of other samples is appropriate which cannot find few voids in other pictures.
9. In p4 line 144, what is IN718? Is that same with Inconel 718 superalloy?
IN718 was short for Inconel 718 superalloy, and they are the same things. We have changed “IN718” to “Inconel 718 superalloy” in p4 line 145.
10. I suppose that the unit of horizontal axis of Fig. 2 is “h”.
We have already changed “min” to “h” in Fig. 2.
11. Please add the unit in Table 3.
We have already added the unit “μm” for the size of δ phase in Table 3, 5, 6 and 7.
12. In p6, the authors mentioned that the Ni content of γ decreased during aging and then δ precipitated. However, I suppose that the opposite phenomenon happened: δ precipitated and then the Ni content of γ decreased.
In p6, we are not mentioned that the Nb content of γ decreased during aging and then δ precipitated. It is right that δ precipitated can consume plenty of Nb in γ phase which can cause the Nb content of γ decrease in our paper.
13. The authors used EDS to measure the Nb content. However, I recommend to use WDS by using EPMA because EDS is not suitable for a quantitative analysis. I recommend to investigate the composition of δ phase in addition to other phases.
Though EDS may cause the measurement result to be not accuracy enough compared with EPMA, it still can let us know the content of main element such as Nb in the inter dendrite, the core dendrite and other large areas, which means that it does not affect our final results.
14. Why did the authors use the sample aged for 12 h for EDS? Both Fig. 1 (a) and Fig. 3(a) is obtained from the sample before aging, but the microstructure looks quite different. Why? In addition, Fig. 3(a), (c) and (e) are images observed from the same region. So, it is better to show measurement point in one image. In Fig. 3(f), the measurement location looks indicate δ phase, not γ phase.
We want to know condition between aging 8 h and 16 h, so we choose the condition at aging 12 h to identify the content of Nb in different areas.
Fig. 1 (a) and Fig. 3(a) have a big difference due to the different positions of samples, which is means that Laves phase in some areas is long strip shaped, and it in other areas is large block shaped. Another reason is Fig. 1 (a) has larger amplification multiplier than that in Fig. 3(a). Therefore, Fig. 1 (a) is only local section of Fig. 3(a).
Though Fig. 3(a), (c) and (e) are images observed from the same region, only one point can be measured in each test in the process of EDS measurement.
Because δ phase is always located in the inter dendrite (closed to Laves phase) and γ (eutectic) area (far away from Laves phase). Therefore, the measurement location is δ phase which is located in γ (eutectic) area in Fig. 3(f).
15. How did the authors judge that the shape of δ phase changed from rod-like to needle-like?
We read more references about the shape of δ phase, and they always define that the length of δ phase is short only formed in the inter dendrite called rod-like δ phase. While the length of δ phase is long grown from the inter dendrite to core dendrite called needle-like δ phase.
16. p8 line 208-210, the authors mention that there is no significant difference between Fig. 1(d) and Fig. 5(d). However, the length, the number of precipitated δ and the number of preferential growth direction of δ in Fig. 5(d) is larger than those in Fig. 1(d). In addition, I suppose that the increase of the number of preferential growth direction of δ phase is one of the big effects of the addition of electromagnetic field while the authors don’t mention about that.
We believe that the number and size of δ phase precipitated in Fig. 5(d) is larger than that in Fig. 1(d). And δ phase still grow in the inter dendrite. Compared Fig. 6(d), the number and size of δ phase precipitated in Fig. 5(d) and in Fig. 1(d). And δ phase grow from the inter dendrite to core dendrite which is full of all picture nearly. Therefore, we get the result that there is no significant difference between Fig. 1(d) and Fig. 5(d), and there is a significant difference between Fig. 1(d) and Fig. 6(d).
Electromagnetic field can not directly affect the growth direction of δ phase, but it can bring an significant influence in the growth rate of δ phase. As we all known, the crystal orientation relationship between δ phase and matrix γ phase is //、// , which means that the direction of δ phase depends on the direction of γ phase. In addition, there is an orientation relationship between δ phase and Laves phase, which is //、//. Therefore, the direction of δ phase growth changes is attributed to the electromagnetic field effect on γ phase and Laves phase.
17. In Fig. 10, the error bars are quite big. I suppose that the load was too small and the authors
measured hardness of a limited region. I recommend to increase the load and measure the hardness from larger region.
Maybe it depends on different equipments. The measuring errors is around ± 50 HV in our equipment.
18. What are VV, AA, LL and PP?
AA is Area Percent, which means that a phase take up a fraction of the total area of the sample in the area of the picture. VV is Volume percentage. LL is Line fraction. PP is Point fraction.
19. For fitting analysis like Fig. 9, it is better to use over four points.
Three points can fit a straight line, which can reflect the trend of line.
20. Why did the time index n change drastically when electromagnetic field intensity was set to 50 mT? Please explain the relationship with the precipitation kinetics.
We can get the changes law about the time index n, which need to apply the Avrami equation.
In the Avrami equation, if we fix Vδ, Vs,δ and t, precipitation rate of δ phase α become so large when electromagnetic field intensity was set to 50 mT. So the time index n should be decreased at the same time, which maintain bilateral equilibrium of equation.
21. Line 326-328 is only about the sample with electromagnetic field of 0 mT?
We have already written that “It could be witnessed that the microhardness of samples were increased with electromagnetic field intensity extended in different aging time as is shown in Fig. 10 and Table 11”, which conclude the electromagnetic field of 30 mT and 50 mT.
22. How much is the hardness before aging treatment?
We have already written the hardness before aging treatment in our paper “original sample whose microhardness reached 300 HV without electromagnetic field intensity and without any heat treatments” .
23.Lastly, there are numerous mistakes and errors in sentence structure in the manuscript. Please consider proof reading either using a proof-reading service or a native speaker with a scientific background. I suppose that “believe” and “someone” are not used in a scientific paper.
We have already changed “believe” to “have studied” and “have reported”.
We have already changed “someone” to “some researches”.

Round 2
Reviewer 3 Report
Over the past month, I took part in a scientific conference and read several articles by other authors on this topic. This topic is really relevant.
I believe that the article has flaws, but it can be published in a new edition.
I accept the answers to my comments.
Author Response
Responds to reviewer 3:
1 I believe that the article has flaws, but it can be published in a new edition.
Thanks for you very much for your time on the manuscript. We have made some correction to the manuscript according to the comments by the 4 reviewers. We think it is can be accepted by the journal for considering publishment.
Reviewer 4 Report
The authors should write what are VV, AA, LL and PP in the manuscript.
Please explain the relationship between time index n and the precipitation kinetics in the manuscript.
The authors replied that the composition of the alloy doesn’t change in the experiment. So, please show the composition of the alloy after laser solid forming process.
There are some mistakes. The authors should check again.
Author Response
Responds to Reviewer 4;
1. The authors should write what are VV, AA, LL and PP in the manuscript.
Thank you for your comment.
We have already given some information about VV, AA, LL and PP in the manuscript, such as the following,
AA is area fraction, which means that a phase take up a fraction of the total area of the sample in the area of the picture. VV, LL, PP are volume fraction, line fraction and point fraction.
2. Please explain the relationship between time index n and the precipitation kinetics in the manuscript.
Thank you for your comment.
We have already explained the relationship between time index n and the precipitation kinetics in the manuscript, such as the following,
Time index n is the slope of the straight line in Fig. 10. It gradually decreased with the increase of the electromagnetic field intensity, and it went down about a half when the electromagnetic field intensity increased from 30 mT to 50 mT. When time index n is smaller with electromagnetic field intensity increasing, the slope of the straight line is also lower, which means that ordinate at the origin (precipitation rate of δ phase α ) become larger. Therefore, plenty of δ phase can precipitate in a shorter time if electromagnetic field intensity is so high.
3. The authors replied that the composition of the alloy doesn’t change in the experiment. So, please show the composition of the alloy after laser solid forming process.
Thank you for your comment.
We have already modified the Table 1 to show the difference between chemical composition of Inconel 718 superalloy powder and laser solid forming(LSF) sample in the manuscript, such as Table 1 in revised manuscript.
It can get the result that the composition of the LSFed alloy basically unchanged after laser solid forming process.
4. There are some mistakes. The authors should check again.
We have correct the manuscript carefully to avoid any mistakes. All the correction has been marked out in the revised manuscript.